# Study of Carbonated Clay-Based Phosphate Geopolymer: Effect of Calcite and Calcination Temperature

Rania Derouiche [1,2], Marwa Zribi [1] and Samir Baklouti [1,*]

[1] Laboratory of Advanced Materials, National School of Engineers of Sfax, University of Sfax, Sfax 3038, Tunisia
[2] Department of Materials and Chemistry, Vrije Universiteit Brussel, Pleinlaan 2, 1050 Brussels, Belgium
* Correspondence: baklouti.samir@gmail.com; Tel.: +216-9723-5972

**Abstract:** This study aims to use natural carbonated Tunisian clay as an aluminosilicate precursor for the elaboration of phosphate-based geopolymers, which yields to the valorization of this common material in Tunisia. In addition, the presence of calcium carbonate in this clay allows the investigation of this associated mineral's effect on the properties of geopolymeric materials. To achieve these purposes, several experimental techniques were used, namely fluorescence (FX), particle size analysis, thermogravimetric analysis (TGA), differential thermal analysis (DTA), dilatometric analysis, X-ray diffraction (XRD), Fourier-transform infrared spectroscopy (FTIR), and scanning electron microscopy (SEM). The mechanical strength and the open porosity of the obtained geopolymeric samples were tested by the compression test and the standard test method for water absorption, respectively. The findings of this work show that the used Tunisian clay can present an attractive aluminosilicate precursor for the synthesis of phosphate-based geopolymers. It also shows that the chosen calcination temperature of the raw clay considerably modifies the reactivity of minerals during geopolymerization and, consequently, strongly affects the properties and structure of the geopolymeric samples. These effects were attributed essentially to the formation of new calcium crystalline phases in the obtained geopolymeric samples. In fact, the anorthite ($CaAl_2Si_2O_8$) phase appears in all the samples but in greater abundance in those obtained from the clay calcined at 550 °C, and the brushite phase ($CaHPO_4 \cdot 2H_2O$) appears only in the samples obtained from the clay calcined at 950 °C. All these new crystalline phases are strongly dependent on the state of the calcite present in the calcined clay.

**Keywords:** carbonated clay; acid-activated; phosphate-based geopolymers; calcination temperature; calcite

## 1. Introduction

For several decades, studies have been undertaken with the aim of developing new materials that consume less energy and are more suitable for the environment. In this context, advanced research has been directed towards a new eco-material named "geopolymer". Since their discovery, the understanding of the synthesis and properties of these materials has greatly improved. In addition, the commercialization of geopolymers on a large scale has become possible. These new materials are generally obtained by alkaline or acid activation of aluminosilicate precursors [1]. Several studies have compared these two different types of activation and claim that geopolymeric materials obtained by acidic activation have superior performance than those obtained by alkali activation [2–4]. In addition, this activation pathway yields phosphate valorization (generally, the acid used is the phosphoric one), which is becoming more and more recommended all over the world. Although acid-based phosphate geopolymers are the most widely reported, the formation of alkali-based geopolymers in which the tetrahedral silicate groups are partially replaced by phosphate groups has also been reported [5].

Several materials can be used as the aluminosilicate precursor. Few studies have been conducted on the utilization of common natural clays in the formulation of geopolymers. In

fact, natural clays are rich in associated minerals compared to kaolin clay, which is simpler in terms of structure and composition. Most of the research on the synthesis of phosphoric acid-based geopolymers used kaolin as a raw material [3,6–10]. However, Louati et al. [11], Douiri et al. [12], and Ben masseoud et al. [13] are among the few researchers who are interested in the synthesis of geopolymers from natural clay and phosphoric acid.

In addition, natural clay materials often constitute complex mixtures of minerals whose particle size and physico-chemical properties are highly variable. Nowadays, the use of clays, in particular those which are rich in associated minerals, is very required, such as natural calcium carbonate with the chemical formula $CaCO_3$ [14]. It contains one of the most common minerals on Earth [15]. Following the thermal treatment, the calcite decarbonation phenomenon can take place between 650 and 860 °C, with the emission of a large quantity of gas ($CO_2$) [16]. In the presence of metakaolin, the formed calcium oxide (CaO) can form an anorthite ($CaAl_2Si_2O_8$) phase [17], which helps to strengthen the mechanical properties of the geopolymeric materials and the formation of the vitreous phase that fills the open porosity [18].

The purpose of making geopolymeric materials from clays rich in calcium carbonate is to benefit from this compound's effect on the composite obtained geopolymer properties [19–21]. This effect has been largely studied with alkaline-based geopolymers [22–27]. However, with the phosphate-based one, it is not yet well studied.

This study aims to use natural Tunisian clay as an aluminosilicate precursor for the elaboration of phosphate-based geopolymeric materials. Such work yields to the valorization of this very abundant material in Tunisia. The selected natural clay is of the illito-kaolinitic type, rich in calcium carbonate. So, we give more attention to the effect of this associated mineral on the synthesis and characteristics of geopolymer materials.

## 2. Materials and Methods

### 2.1. Materials

The two reagents used for the synthesis of phosphate-based geopolymeric materials in this study are local natural clay and phosphoric acid. The natural clay, rated A25, comes from the central region of Tunisia and represents the aluminosilicate precursor chosen for the synthesis of geopolymer materials. According to the X-ray fluorescence analysis presented in Table 1, this clay is characterized by a chemical composition rich in silicon dioxide ($SiO_2$ = 56.96 wt%) and aluminum oxide ($Al_2O_3$ = 16.77 wt%). Thus, the Si/Al molar ratio is about 2.94, which probably indicates the presence of free silica and/or minerals of 2:1 clay type as illite, as can be seen in the XRD result (Figure 1). This is supported by the presence of potassium oxide ($K_2O$) with a content of 3.24 wt%. In addition, the high oxide content of iron ($Fe_2O_3$ = 6.675 wt%) and the lower content of magnesium oxide (MgO = 2.230 wt%) may suggest the presence of clinochlore and/or dolomite and hematite or goethite. Finally, this clay contains a significant percentage of calcium oxide (CaO = 5.32 wt%). This high grade reflects the presence of calcite and/or dolomite.

**Table 1.** Chemical composition (percentage weight: wt%) of the natural clay (A25).

| Oxides | $Fe_2O_3$ | $Al_2O_3$ | $TiO_2$ | CaO | $SiO_2$ | MgO | $K_2O$ | $Mn_2O_3$ | ZnO | LOI * |
|--------|-----------|-----------|---------|-----|---------|-----|--------|-----------|-----|-------|
| **A25** | 6.67 | 16.77 | 0.84 | 5.32 | 56.96 | 2.23 | 3.24 | 0.05 | 0.02 | 6.51 |

\* LOI: loss on ignition at 1000 °C.

The X-ray powder diffraction (XRD) of A25 clay shows that it is essentially composed of kaolinite ($Al_2O_3.2SiO_2.2H_2O$), illite ($KAl_2(Si_3Al)O_{10}(OH)_2$), clinochlore ($Mg_{3.75}Fe^{2+}_{1.25}Si_3Al_2O_{10}(OH)_8$), calcite ($CaCO_3$), and quartz ($SiO_2$) (Figure 1). In this clay, the presence of different minerals with considerable percentages attracts our attention to study their interaction with phosphoric acid in the elaboration of phosphate-based geopolymeric materials and their effects on the obtained material structure and properties.

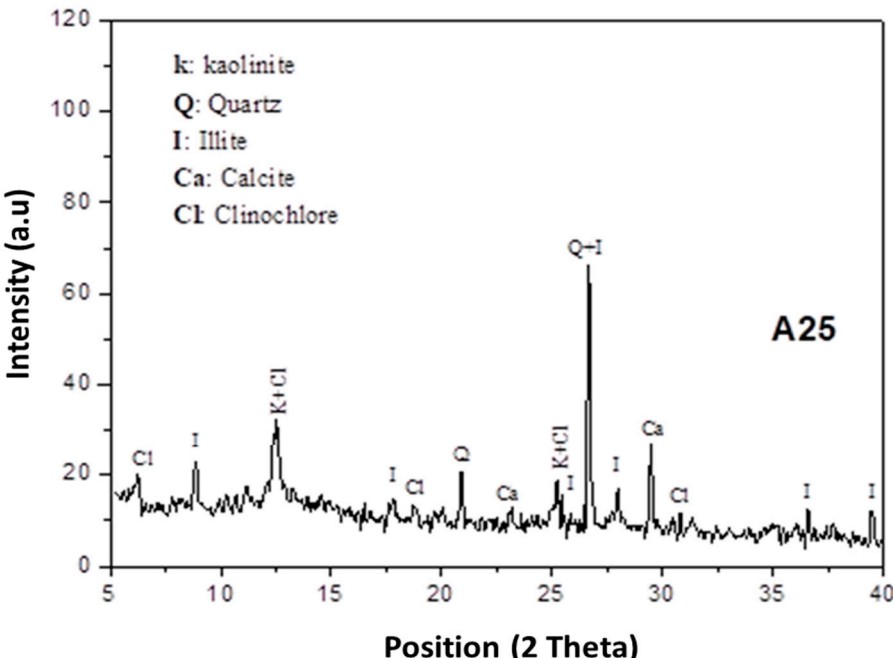

**Figure 1.** Diffractograms of A25 sample with K—Kaolinite (00-003-0058), Q—Quartz (01-082-0511), I—Illite (00-026-0911), Ca—Calcite (00-002-0629), Cl—Clinochlore (00-029-0701).

This clay was crushed and then ground using a planetary grinder. Then, it was sieved through an 80 μm sieve and heated under an oxidizing atmosphere at different temperatures: 550 °C, 750 °C, and 950 °C, with a rate of 5 °C/min for 3 h. The choice of these specific calcination temperatures was made according to the DTA/TGA analysis of the raw clay and its mineralogical composition after each thermal treatment. More details are presented in Section 3.1.2, which will be reserved for the discussion of this experimental technique's results. The heated clay powder was again sieved at 80 μm to avoid the agglomerates formed during the calcination and to ensure the fineness of the particles. The second reagent used is the phosphoric acid solution, which presents the activating solution for the synthesis of geopolymer materials. This acid is characterized by a concentration of 85% $H_3PO_4$ and is supplied by the company Scharlau-chémie SA. Finally, distilled water was used to prepare the different mixtures and to maintain a constant water/powder mass ratio (W/P = 0.4).

*2.2. Samples Preparation*

The protocol adopted for the synthesis of geopolymeric materials is divided into two stages: The first stage deals with the preparation of the activating acid solution by mixing adequate quantities of acid and water to obtain the sustainable and optimal P/Al molar ratio (where Al is the number of moles of aluminum coming from the source of aluminosilicate and P is the number of moles of phosphorus coming from the phosphoric acid). Following Zribi et al. [28] and in function of the results of preliminary tests, this ratio was fixed equal to 1. The second step of the synthesis protocol consists of adding the aluminosilicate precursor to the acidic solution. Then, the well-homogenized and obtained paste is placed in a cylindrical plastic mold with a diameter of 27 mm and kept in an oven at 60 °C for 24 h. After curing, the cylindrical samples (in the form of pellets with a diameter of 27 mm and a thickness of around a quarter of the diameter) are kept at room temperature for 28 days before being characterized by different techniques. These conditions were chosen based on previous studies reported by Derouiche et al. [10].

The samples were labeled as AX, $GA_1{}^X$, and $GK_1{}^X$, where:

X refers to the corresponding temperature.

A alludes to natural clay.

G refers to geopolymeric materials.

K denotes calcined kaolin.

Number 1 represents the P/Al molar ratio.

### 2.3. Methods

The chemical composition of untreated and treated clay was determined by X-ray fluorescence using a Philips X'unique II instrument. The particle size distributions of untreated and treated clay were measured by laser diffraction (Fritsch analyzer 22 microtec plus). The differential thermal analysis/thermogravimetry (DTA/TGA) analysis was made using Setaram SETSYS Evolution-1750 with a 10 °C/min heating rate from room temperature up to 1200 °C under nitrogen flux. Measurements were carried out on approximately 20 mg of sample mass. The mineral phases were identified by X-ray diffraction measurements of different specimens using a "BRUKER-AXS-D8" type powder diffractometer with CuK$\alpha$ radiation ($\lambda$k$\alpha$ = 1.5418 Å). A 2$\theta$ angle interval of 5 ° to 70 ° was scanned with a speed of 1°/min. High Score Plus software was used to identify the crystallographic phases and to determine the integral intensity of an individual diffraction peak. The Fourier-transform infrared spectroscopy (FTIR) was scanned from 400 to 4000 cm$^{-1}$ wavenumbers with a resolution of 2 cm$^{-1}$ using Perkin Elmer spectrum BX spectrophotometer apparatus in transmittance mode. The analysis was carried out on KBr pellets. The Brazilian tests were conducted on geopolymer pellets of 27 mm in diameter and 6 mm in height to measure the diametric compressive strength of phosphoric acid-based geopolymers using a LLOYD LR 50K PLUS universal testing machine with a crosshead speed of 0.2 mm/min. All values of the mechanical strength were equal to the average of the measurement over five tests under the same conditions and were expressed in MPa. Scanning electron microscopy (SEM) was performed using a thermo scientific microscope on broken samples of geopolymeric samples at an accelerating voltage of 10 kV. The fragments were not polished or coated. The open porosity of the geopolymers was determined using the water absorption method. The samples were dried at 100 °C, later immersed in water at room temperature for 24 h, and weighed afterward. The difference between the dry weight and the weight after removing them from the water provided the weight of water in the open pores. Mechanical strength and porosity were measured each time for three specimens and averaged. A summary of the flow sheet of experimental protocol and methods adopted is exhibited in Figure 2.

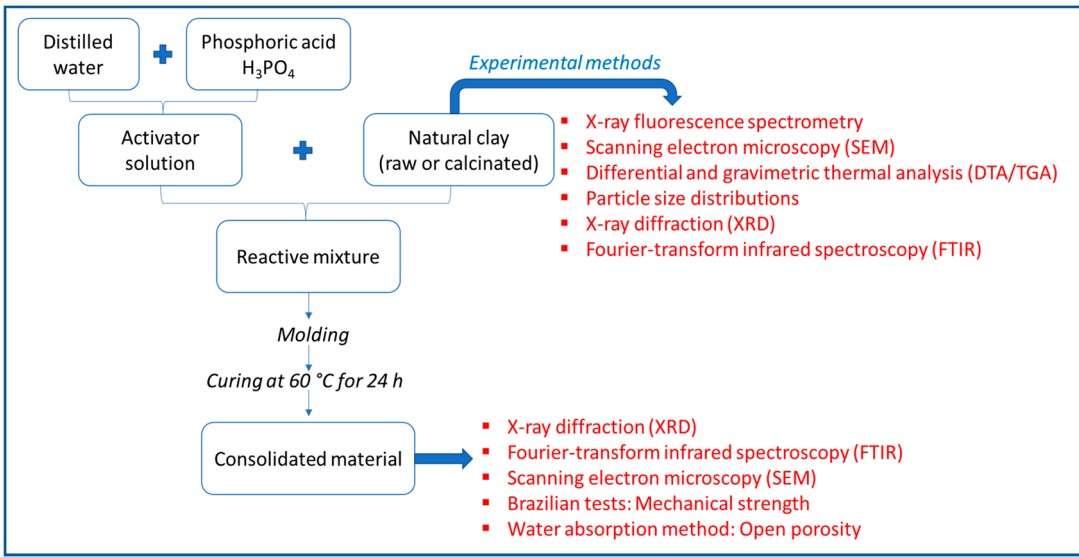

**Figure 2.** The flow sheet of experimental protocol and methods adopted.

## 3. Results

### 3.1. Characterization of Raw and Calcined Tunisian Clay

3.1.1. SEM Analysis of Raw Clay

Figure 3 shows the raw clay-based sample microstructure. According to this shot, the A25 clay consisted mainly of fine particles in the form of flat platelets, sometimes in the form of hexagonal, of different sizes, oriented in random directions [15]. This morphology and the particles' fineness were generally characteristic of clay minerals, such as kaolinite and illite [29], which probably predominate the composition of this clay.

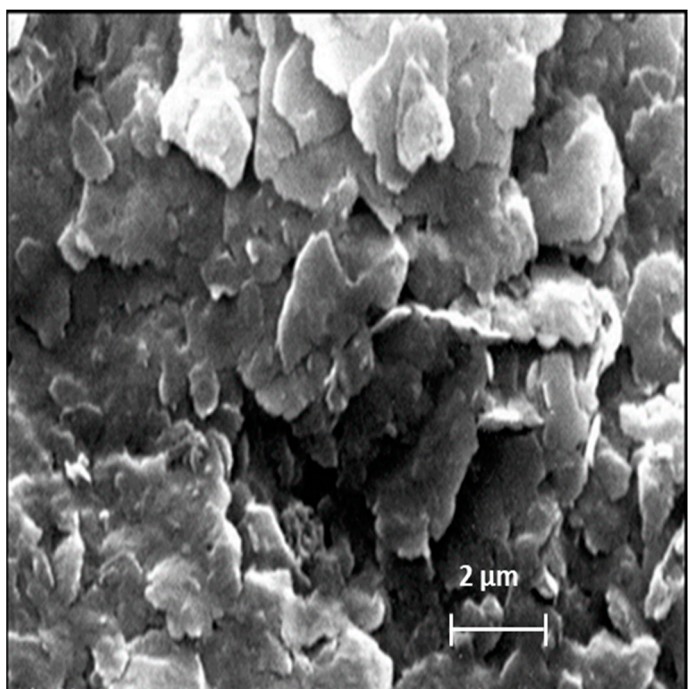

**Figure 3.** SEM analysis of raw clay.

3.1.2. Thermal Behavior of Raw Clay

The DTA/TGA thermograms of the clay A25, presented in Figure 4, show several peaks relating to thermal phenomena accompanied sometimes by mass variation. Two endothermic peaks located at 150 °C and 200 °C were accompanied by a loss in mass of 3.5% of the initial mass of the sample. These peaks are related to the release of free water (physiosorbed water) and zeolitic water (interfoliar water) [30]. In addition, an endothermic fluctuation around 300 °C, accompanied by a loss in mass of 0.5%, can be attributed to the dehydroxylation of geothite ($FeOOH \rightarrow Fe_2O_3 + H_2O$) [31–33], which may be present and responsible for the high $Fe_2O_3$ content. The release of the water of constitution linked to the dehydroxylation of kaolinite and its transformation into metakaolin is characterized by an endothermic peak. This peak's maximum is observed at 533 °C and confirmed by a loss in mass of 2.9%. This mass loss is lower than that expected for pure kaolinite, which is around 14% [34], which confirms the presence of kaolinite in a low amount and of other associated minerals. In addition, another endothermic phenomenon of low intensity and without mass change, observed around 573 °C, is characteristic of the allotropic transition of α-quartz to β-quartz [35]. Finally, a last endothermic peak located at 750 °C corresponds to the decomposition of the calcite with the release of $CO_2$ [16]. This technique confirms once again the presence of calcite in the A25 clay and allows one to identify its decomposition domain. The absence of the exothermic peak around 950 °C, corresponding to the transformation of metakaolinite into mullite, may be due to the low content of kaolinite and the presence of impurities such as iron oxide and calcium carbonate in the A25 clay.

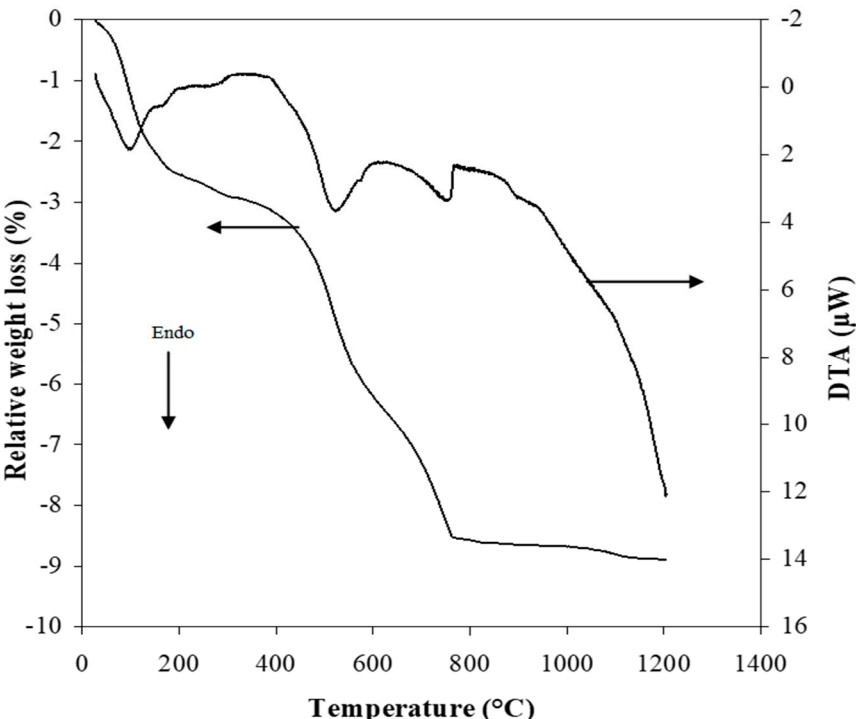

**Figure 4.** DTA/TGA curves of A25 sample.

Based on these results, the following calcination temperatures were chosen:
- T = 550 °C: the temperature at which only the dehydroxylation of the kaolinite has taken place [36], and all the calcium carbonate is present.
- T = 750 °C: the temperature at which the calcium carbonate is decomposed [16,30].
- T = 950 °C: the temperature at which the dehydroxylation of illite is dehydroxylated [37–39].

These three temperatures modify in a notable way the phases present in the calcined clay and, consequently, its reactivity towards phosphoric acid, allowing us to investigate the effect of the presence of calcite in the base clay. These thermal transformations are correlated with the result reported in Section 3.1.4, which will be reserved for the discussion of the XRD analysis of raw and calcined clay at 550 °C, 750 °C, and 950 °C.

### 3.1.3. Particle Size Distribution of Raw and Calcined Clays at 550 °C, 750 °C, and 950 °C

The particle size distribution of raw clay (A25) and calcined ones (A550, A750, and A950) was determined by laser particle sizing, and the obtained results are shown in Figure 5. The treatment of the granulometric distribution results shows that the size of the particles increases with the increase in the temperature of calcination, except at 950 °C, where a decrease is observed. From room temperature to 750 °C, the curves show a bimodal granulometric distribution, thus indicating the presence of two populations: one relating to the finest clay minerals (centered at approximately 2 μm at 25 °C) and the other centered at approximately 30 μm at 25 °C due to the presence of other phases such as quartz, carbonates, and other associated minerals [40]. The shift of the granulometric distributions of the calcined clay up to 750 °C indicates an increase in the size of the grains following an agglomeration phenomenon. On the other hand, the clay calcined at 950 °C has a single population and a mono-modal distribution. This can be explained by the total disappearance of the fine fraction in favor of the coarse fraction. This behavior can also be justified by the beginning of the sintering phenomenon at this temperature which decreases the size of the formed agglomerates.

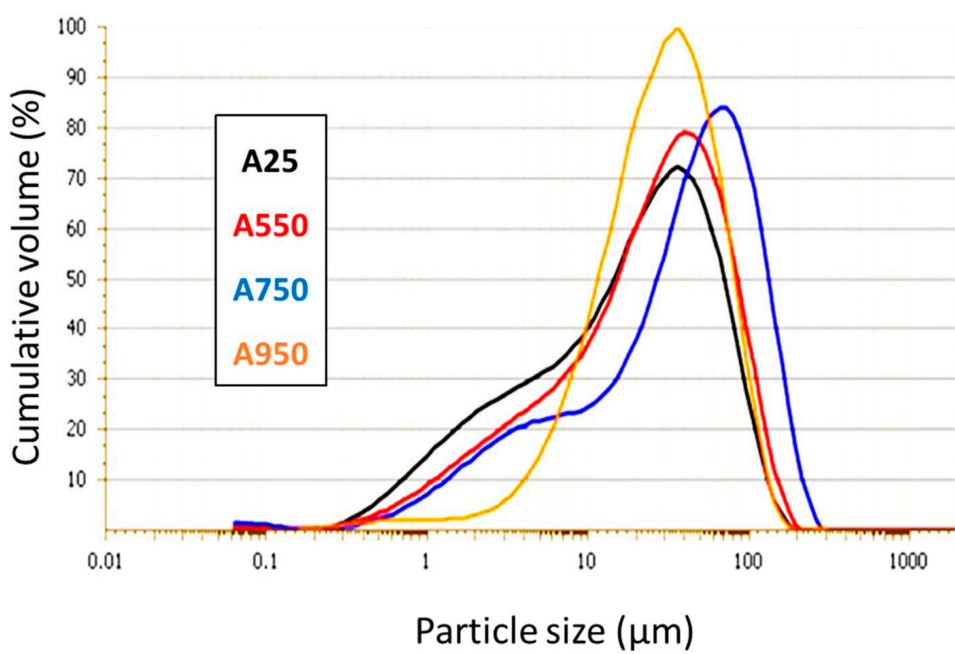

**Figure 5.** Particle size distribution of A25, A550, A750, and A950 samples.

3.1.4. XRD Analysis of Raw and Calcined Clay at 550 °C, 750 °C and 950 °C

The results of X-ray diffraction analysis of A25 raw clay and calcined ones at different temperatures (550 °C, 750 °C, and 950 °C) are shown in Figure 6.

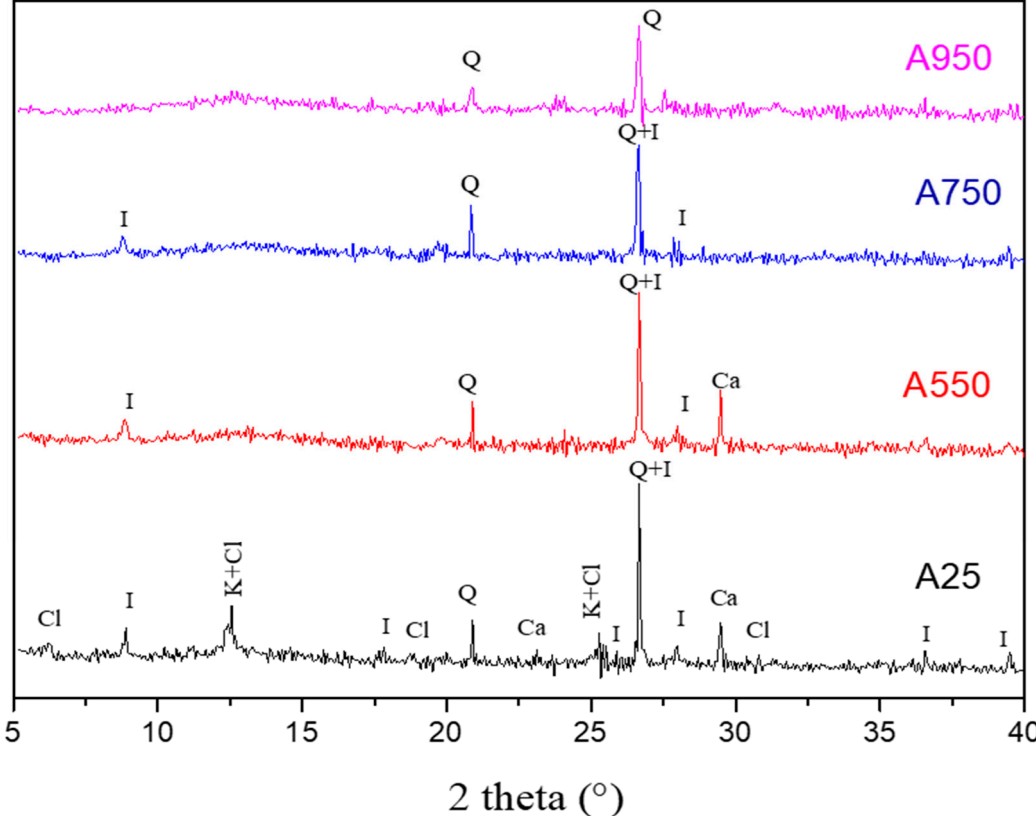

**Figure 6.** Diffractograms of A25, A550, A750, and A950 samples with K—Kaolinite (00-003-0058), Q—Quartz (01-082-0511), I—Illite (00-026-0911), Ca—Calcite (00-002-0629), Cl—Clinochlore (00-029-0701).

The X-ray diffraction spectrum shows that the A25 clay is essentially composed of two clay minerals, namely kaolinite ($Al_2O_3.2SiO_2.2H_2O$) and illite ($KAl_2 (Si_3Al) O_{10}(OH)_2$). In addition, this diffractogram allows us to highlight the presence of characteristic lines of clinochlore ($Mg_{3.75}Fe^{2+}_{1.25}Si_3Al_2O_{10}(OH)_8$), calcite ($CaCO_3$) and quartz ($SiO_2$), which is characterized by the most intense peak ($2θ = 26.73$). Thus, the studied A25 clay presents an illito-kaolinitic clay rich in calcium carbonate [41]. The analysis of the diffractograms of the calcined clay shows that several characteristic lines of the various identified minerals have disappeared. Indeed, from 550 °C, all the characteristic peaks of clinochlore disappear. At this same temperature, the absence of the characteristic lines of kaolinite, following the dehydroxylation phenomenon, confirms the formation of metakaolinite [42]. This is also proved by the appearance of an amorphous phase identified by a dome located between ($2θ = 11°$ and $2θ = 15°$). When the temperature reaches 750 °C, we note a disappearance of the calcite peak following the phenomenon of decarbonation. The diffractogram of the calcined clay at 950 °C indicates the total disappearance of the illite peaks and the persistence of the quartz crystal structure.

Finally, it is worth noting that the XRD analyses of raw and calcined clay are in agreement with the previous techniques' results related to the composition of the studied clay after each thermal treatment.

### 3.1.5. FTIR Spectra of the Raw and Calcined Clay at 550 °C, 750 °C, and 950 °C

The FTIR spectra corresponding to the raw and calcined clay at different temperatures (550 °C, 750 °C, and 950 °C), in a spectral range between 500 cm$^{-1}$ and 4000 cm$^{-1}$, are shown in Figure 7.

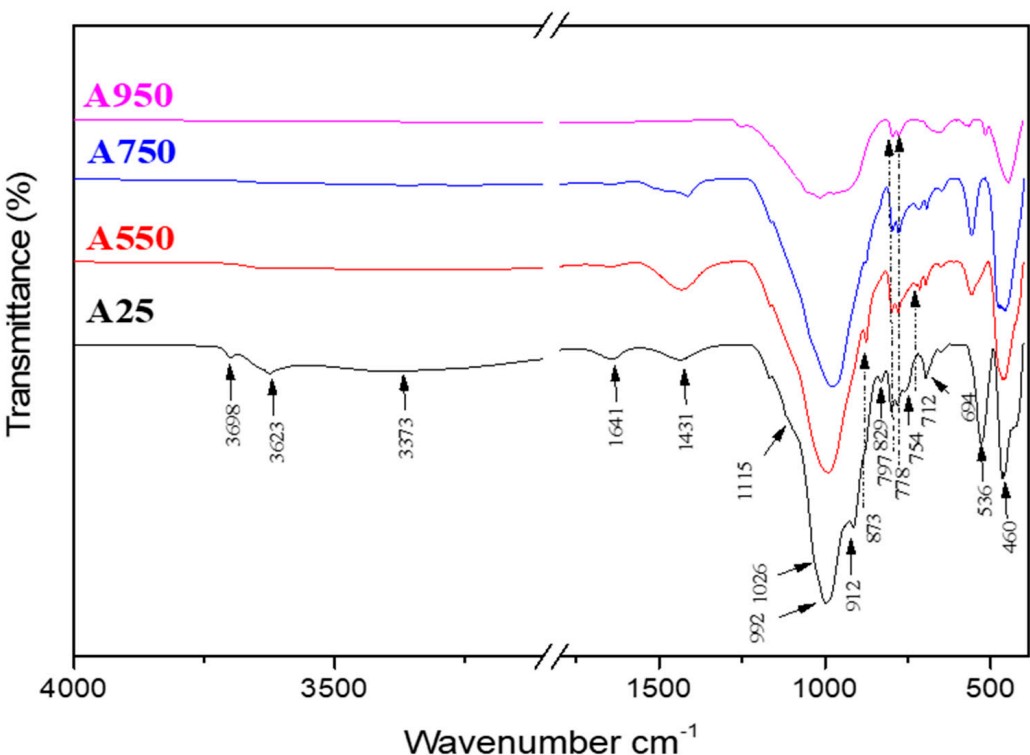

**Figure 7.** Infrared spectra of A25 A550, A750, and A950 samples.

The comparative study of the FTIR spectra of the Tunisian clay before and after calcination at different temperatures shows that several changes take place following the thermal treatment of the clay. Indeed, the calcination at 550 °C causes the disappearance of OH hydroxyl groups related to free and constitutive water molecule vibrations (1641 cm$^{-1}$, 3373 cm$^{-1}$). Additionally, we can conclude the disappearance of characteristic kaolinite bands, such as OH hydroxyl bands located between 3698 cm$^{-1}$ and 3623 cm$^{-1}$, Si–O–Al

bands (754 cm$^{-1}$) and Al–OH ones (912 cm$^{-1}$), in comparison with the spectrum of the raw clay. From 750 °C, the bands of vibration of calcite (712 cm$^{-1}$, 873 cm$^{-1}$, and 1431 cm$^{-1}$) begin to disappear following the reaction of decarbonation [30], while the bands located at 829 cm$^{-1}$ and 778–797 cm$^{-1}$ highlight the persistence of illite and quartz, respectively [43]. On the other hand, the characteristic bands of the ν(Si–O) valence bonds located between 992 cm$^{-1}$ and 1115 cm$^{-1}$ seem to be transformed into a single wider one [44]. Thus, the change in the band environment and the disappearance of the Si–O–Al band of kaolinite can indicate the formation of metakaolinite [45].

This technique confirms once again that the calcination of the A25 clay at different temperatures causes the decomposition of several minerals and the formation of new phases, which can act and influence the reactivity of this clay and, consequently, the geopolymerization reaction in the presence of phosphoric acid.

### 3.2. Characterization of Geopolymeric Materials: GA$_1^{550}$, GA$_1^{750}$, and GA$_1^{950}$

3.2.1. XRD Analyses of Geopolymeric Materials: GA$_1^{550}$, GA$_1^{750}$, and GA$_1^{950}$

The X-ray diffractograms of the geopolymeric materials synthesized from calcined clay at different temperatures (550 °C, 750 °C, 950 °C) are respectively illustrated in Figures 8–11. In these figures, the diffractograms of the corresponding calcined clays are also presented in order to facilitate the interpretation of the results.

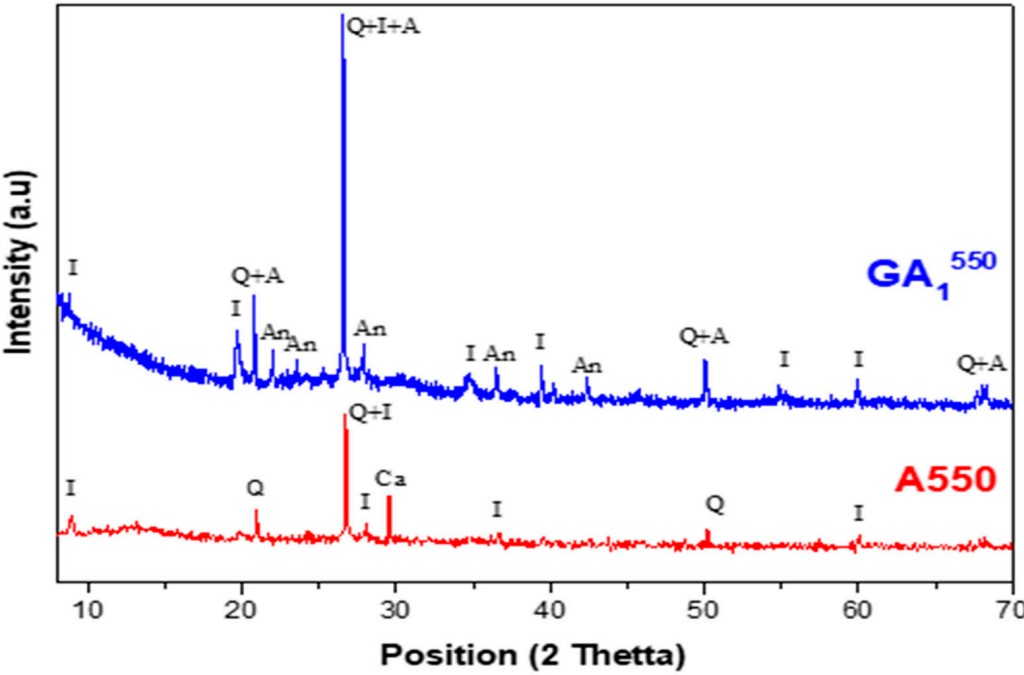

**Figure 8.** Diffractograms of the GA$_1^{550}$ aged 7 days and A550 clay with Q—Quartz (01-082-0511), I—Illite (00-026-0911), Ca—Calcite (00-002-0629), A—Aluminum phosphate (01-084-0853), An—Anorthite (00-012-0301).

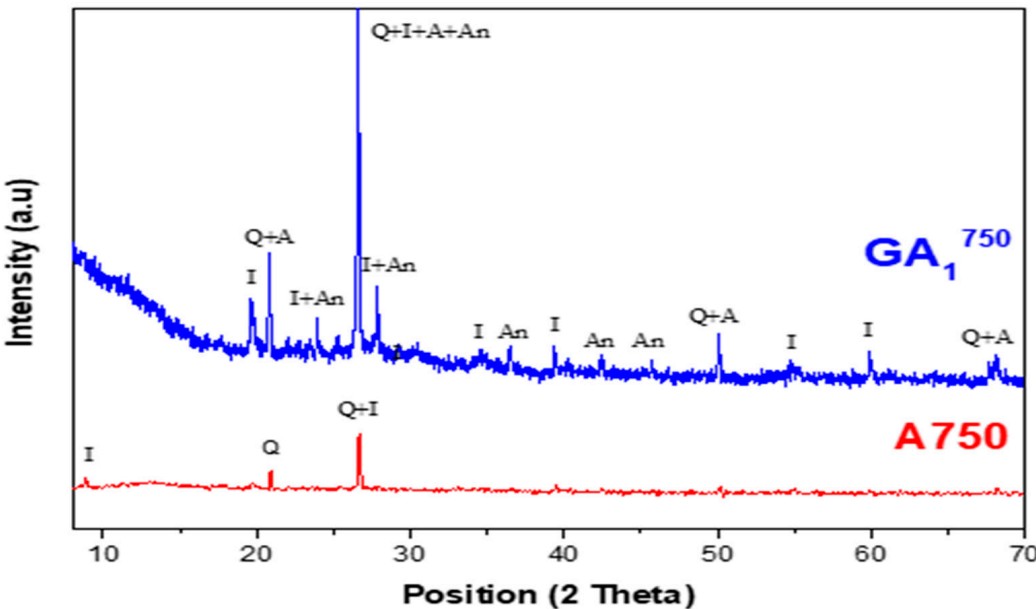

**Figure 9.** Diffractograms of GA$_1^{750}$ geopolymer aged 7 days and A750 clay with Q—Quartz (01-082-0511), I—Illite (00-026-0911), A—Aluminum phosphate (01-084-0853), An—Anorthite (00-012-0301).

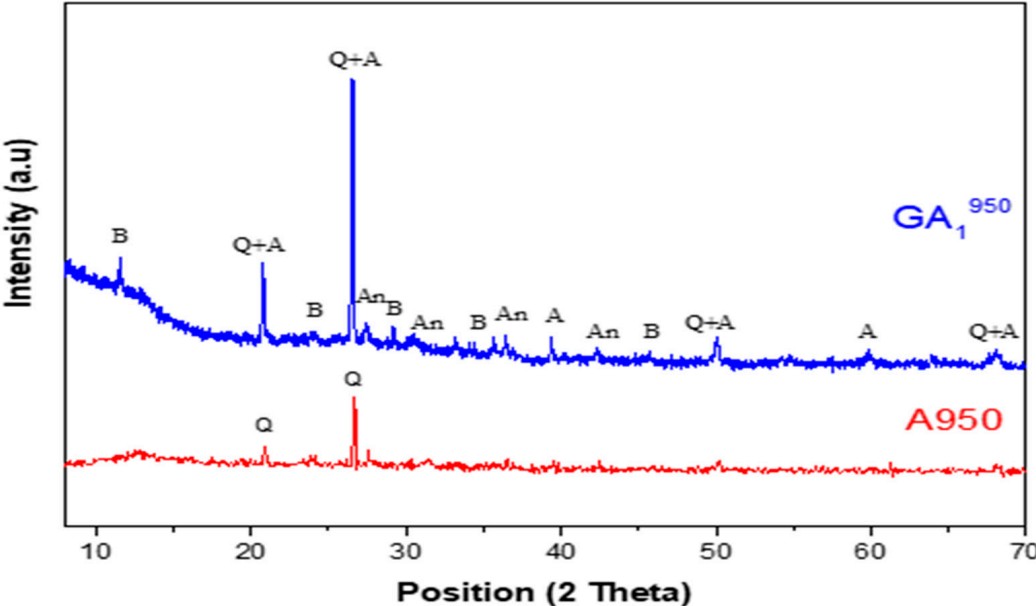

**Figure 10.** Diffractograms of GA$_1^{950}$ geopolymer aged 7 days and A950 clay with Q—Quartz (01-082-0511), I—Illite (00-026-0911), A—Aluminum phosphate (01-084-0853), An—Anorthite (00-012-0301), B—Brushite (00-011-0293).

Comparing the obtained geopolymeric sample with its starting corresponding clay (A550), the diffractogram of the GA$_1^{550}$ geopolymer, presented in Figure 8, shows the following:

- The appearance of a relatively important dome between 2θ = 20° and 2θ = 35°. This dome shows the formation of an amorphous geopolymeric phase in the obtained GA$_1^{550}$ material.
- The appearance of a crystalline aluminum phosphate phase.
- The disappearance of the crystalline phase related to calcite.
- The appearance of a relatively important crystalline phase named anorthite.
- The persistence of the crystalline phases of quartz and illite [46].

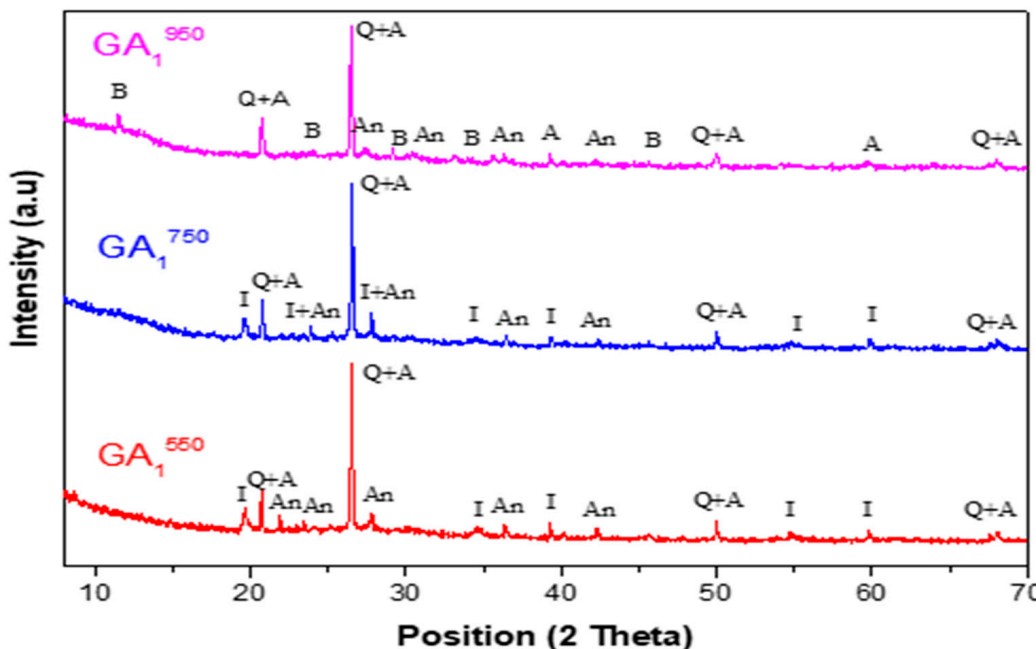

**Figure 11.** Diffractograms of geopolymers synthesized from Tunisian clay calcined at different temperatures (GA$_1^{550}$, GA$_1^{750}$, GA$_1^{950}$) aged 7 days.

These results are well expected if we consider the phases present in the clay calcined at 550 °C. Indeed, the illite and quartz phases are known for their chemical inertia toward phosphoric acid [43]. Thus, they do not contribute to the geopolymerization reaction, and consequently, they are found in the obtained geopolymeric material. The disappearance of the calcite phase in the obtained geopolymeric material proves that this compound was attacked by phosphoric acid. In fact, it decomposed (observation of an effervescence phenomenon during the synthesis of this geopolymer), and then it reacted with the aluminosilicates present (product of the decomposition of clinochlore, for example) to form the anorthite phase (CaAl$_2$Si$_2$O$_8$). The amorphous geopolymeric phase that appeared, as well as the aluminum phosphate, probably present the products of the geopolymerization reaction between the disordered aluminosilicate phase formed (metakaolin and others) and phosphoric acid [47].

The comparison between the diffractograms of the GA$_1^{750}$ geopolymer and A750 clay (presented in Figure 9) shows the following:

- The appearance of a dome between 2θ = 20° and 2θ = 35° is less important than that observed in the diffractogram of the GA$_1^{550}$ geopolymer, as shown in Figure 11. This dome also shows the formation of an amorphous geopolymeric phase in the obtained GA$_1^{750}$ material but with a lower content. This difference can be explained by the decrease in the quantity of the main precursor of the geopolymerization reaction, which is metakaolin. Indeed, it seems that a part of this latter reacts with the lime CaO, formed following the total decarbonation of the calcite at 750 °C, to form calcic aluminosilicate species.
- The appearance of a crystalline phase of aluminum phosphate, which is one of the products of the geopolymerization reaction.
- The appearance of a crystalline phase of anorthite, which is explained by the reaction between the lime and a part of the aluminosilicates phase.
- The persistence of the crystalline phases of quartz and illite. These last ones are indeed inert to phosphoric acid.

The diffractogram of the GA$_1^{950}$ geopolymer, presented in Figure 10, shows the following by comparing it with the corresponding used clay:

- The appearance of a dome between 2θ = 20° and 2θ = 35° is less important than that observed in the diffractogram of the $GA_1^{550}$ geopolymer but slightly more important than that observed in the diffractogram of the $GA_1^{750}$ geopolymer, as shown in Figure 11. This dome shows again the formation of an amorphous geopolymeric phase in the $GA_1^{950}$ material. This difference can be explained on the one hand by the decrease in the quantity of the main precursor of the geopolymerization reaction, which is metakaolin (due to the reaction between a part of metakaolin and the formed lime) and on the other hand by the contribution to the geopolymerization reaction of illite, which undergoes dehydroxylation at about 950 °C (disappearance of the characteristic lines of illite in the diffractogram of the clay calcined at 950 °C A950).
- The appearance of a crystalline aluminum phosphate phase, which is one of the products of the geopolymerization reaction.
- The appearance of a crystalline phase of anorthite, which is explained by the reaction between lime and part of the aluminosilicate phases present. The quantity of this phase seems to be lower than that formed in the $GA_1^{750}$ geopolymer. Indeed, this decrease is justified by the formation of a new phosphate phase, which is brushite ($CaHPO_4 \cdot 2H_2O$).
- The persistence of the quartz crystalline phase.

Table 2 gathers the main crystalline phases presented in the raw and calcined Tunisian clay and in the formed geopolymeric materials.

**Table 2.** A summary of the crystalline phases present in the different compounds.

| Compound | A25 | A550 | A750 | A950 | $GA_1^{550}$ | $GA_1^{750}$ | $GA_1^{950}$ |
|---|---|---|---|---|---|---|---|
| Kaolinite | X | | | | | | |
| Quartz | X | X | X | X | X | X | X |
| Illite | X | X | X | | X | X | |
| Calcite | X | X | | | | | |
| Clinochlore | X | | | | | | |
| Anorthite | | | | | X | X | X |
| Aluminum phosphate | | | | | X | X | X |
| Brushite | | | | | | | X |

### 3.2.2. FTIR Spectra of Geopolymeric Materials: $GA_1^{550}$, $GA_1^{750}$, and $GA_1^{950}$

In order to determine the effect of the calcination temperature of the starting clay and of the clay mineral composition after each heat treatment on the structure of the obtained geopolymer, the geopolymeric materials were also analyzed by the IR spectroscopy technique. Figures 12–14 show the spectra related to the geopolymeric materials synthesized from the calcined clay at different temperatures (550 °C, 750 °C, 950 °C) as well as the corresponding calcined clays.

The FTIR analysis of the $GA_1^{550}$ geopolymer confirms the results obtained by XRD. Indeed, the spectrum of the $GA_1^{550}$ geopolymer, presented in Figure 12, shows the persistence of the bands located at 694 cm$^{-1}$, 778 cm$^{-1}$, and 797 cm$^{-1}$, which are the characteristic bands of the bond (Si–O–Si) of quartz and illite. Thus, we can note the chemical inertia of these phases with respect to the phosphoric acid at a temperature of calcination, which is equal to 550 °C [38,39]. Moreover, we notice the disappearance of the characteristic bands of calcite located at (712 cm$^{-1}$, 873 cm$^{-1}$, and 1431 cm$^{-1}$) of the spectrum of the geopolymer $GA_1^{550}$ by comparing it with that of the calcined clay A550. This can confirm the results observed by XRD regarding the decomposition of this compound under the effect of temperature and phosphoric acid.

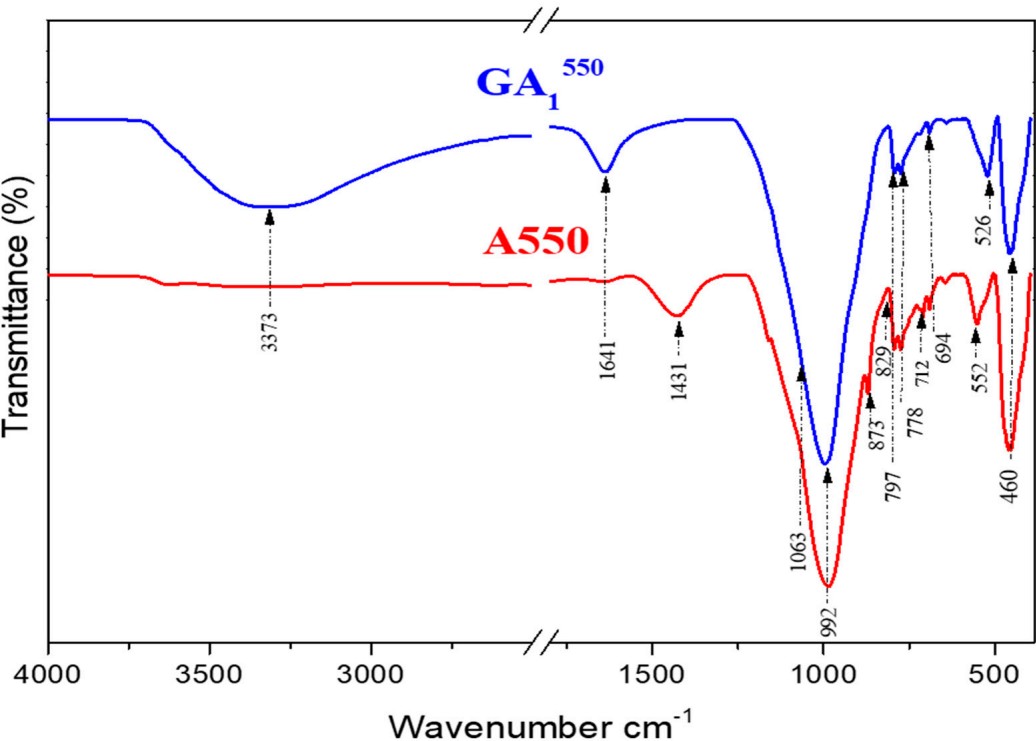

**Figure 12.** Infrared spectra of GA$_1^{550}$ geopolymer aged 7 days and A550 clay.

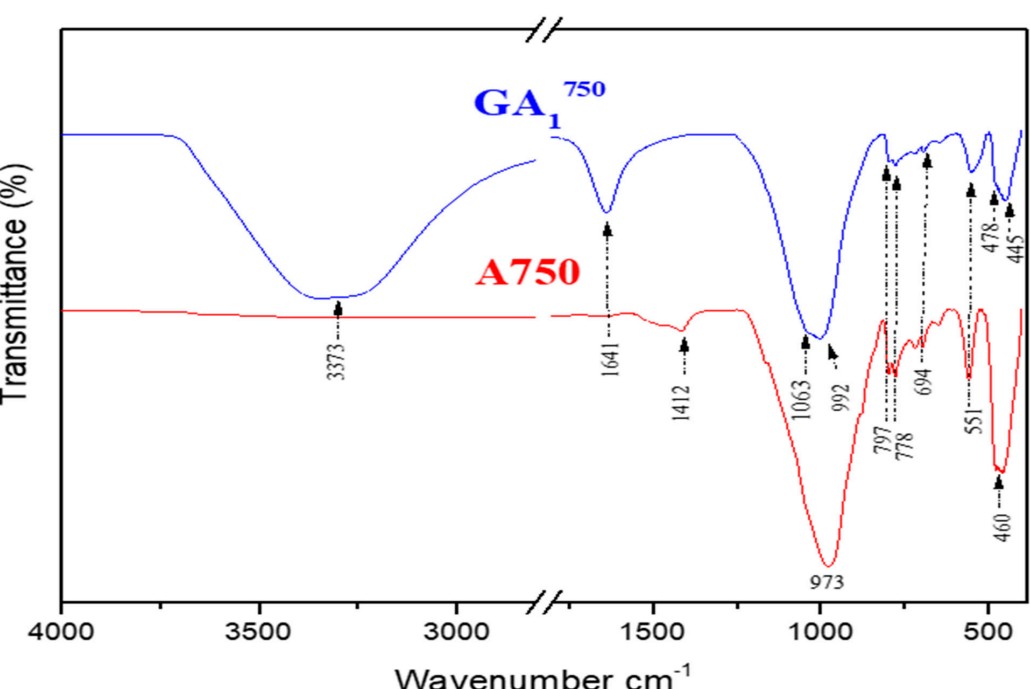

**Figure 13.** Infrared spectra of GA$_1^{750}$ geopolymer aged 7 days and A750 clay.

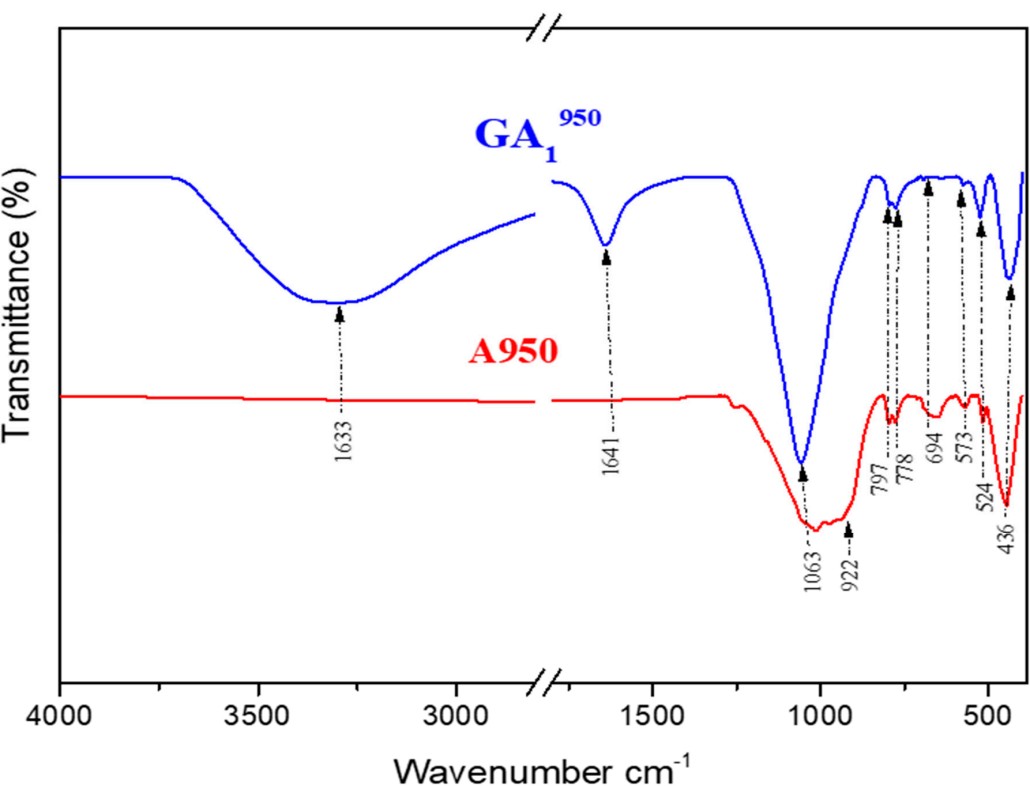

**Figure 14.** Infrared spectra of $GA_1^{950}$ geopolymer aged 7 days and A950 clay.

The analysis of this spectrum also shows the appearance of a band located at (526 cm$^{-1}$) attributed to anorthite ($CaAl_2Si_2O_8$) [40], already observed by XRD analyses. This compound is probably formed during the reaction of calcite with aluminosilicates, which are present in the starting clay. By examining the spectrum of the $GA_1^{550}$ geopolymer, we can also notice that the broad band located at 992 cm$^{-1}$ becomes even broader, and a shoulder towards 1063 cm$^{-1}$ starts to appear. This can be explained by the beginning of the formation of alumina-phosphate compounds from the reaction between metakaolin and phosphoric acid [11].

The last two bands observed around 3373 cm$^{-1}$ and 1641 cm$^{-1}$ are attributed respectively to the modes of vibrations, elongation, and deformation of the groups (O–H) coming from phosphoric acid [48] and to the vibrations of water molecules (H–O–H).

The spectrum of the $GA_1^{750}$ geopolymer compared with that of the clay calcined at 750 °C (Figure 13) shows the following:

The shoulder observed at 1063 cm$^{-1}$ for $GA_1^{550}$ was further developed to give a more intense and brighter shoulder in the $GA_1^{750}$ spectrum, as shown in Figure 15. This can be explained by the increase in the concentration of $GA_1^{750}$ in aluminum phosphate compounds. Indeed, by increasing from 550 °C to 750 °C, more reactive metakaolin is formed, which is the main precursor in the geopolymerization reaction to form new compounds. This is further justified by the appearance of a band located at (478 cm$^{-1}$), which is attributed to the vibration mode of $PO_4$ groups observed in the aluminum phosphate [49]. In addition, the spectrum of $GA_1^{750}$ shows the appearance of new bands located around (445 cm$^{-1}$ and 551 cm$^{-1}$), which are attributed to an anorthite compound [50]. Thus, this phase is increasingly favored by increasing the calcination temperature from 550 °C to 750 °C, which can be clearly observed in Figure 15. The characteristic bands (694 cm$^{-1}$, 778 cm$^{-1}$, and 797 cm$^{-1}$) of quartz and illite persist. These phases still seem to be inert to phosphoric acid, even at 750 °C. Moreover, we observe the persistence of the two bands observed towards 3373 cm$^{-1}$ and 1641 cm$^{-1}$; they are attributed to the modes of vibrations, elongation, and deformation of the groups (O–H) coming from the phosphoric acid [48] and to the vibrations of the water molecules (H–O–H), respectively.

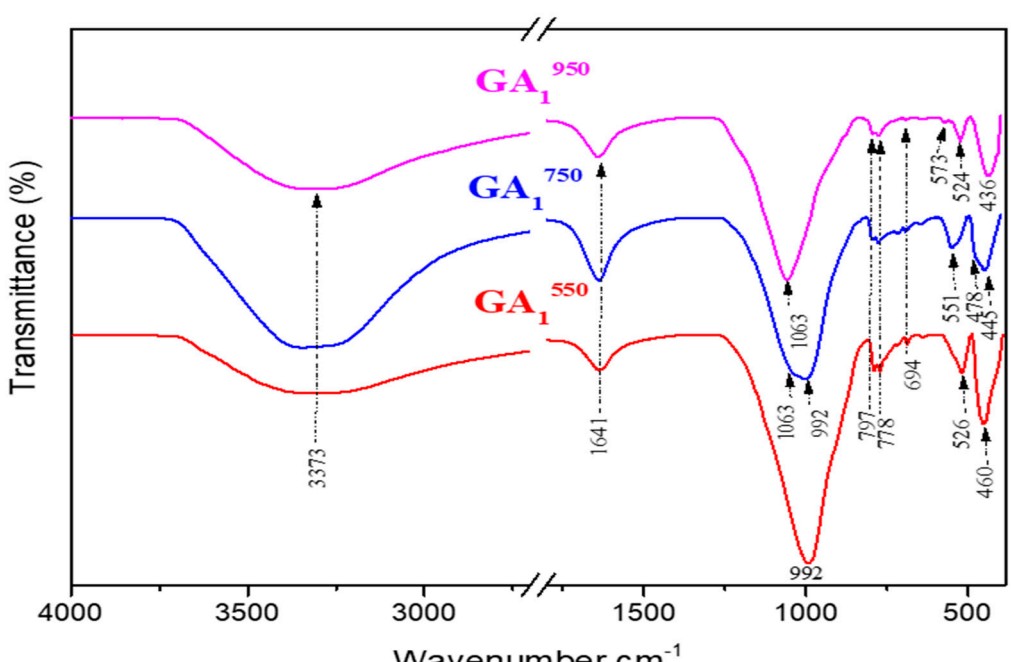

**Figure 15.** Infrared spectra of geopolymers synthesized from Tunisian clay calcined at different temperatures (GA$_1$$^{550}$, GA$_1$$^{750}$, GA$_1$$^{950}$) aged 7 days.

Comparing the spectrum of the GA$_1$$^{950}$ geopolymer presented in Figure 14 with that of the clay calcined at 950 °C, we observe the persistence of the characteristic bands of quartz (694 cm$^{-1}$, 778 cm$^{-1}$, and 797 cm$^{-1}$). One also notices the appearance of a crystalline phase of aluminum phosphate at (436 cm$^{-1}$) which is one of the products of the geopolymerization reaction. Figure 15 proves that the spectrum of the GA$_1$$^{950}$ geopolymer is similar to that of the GA$_1$$^{750}$ geopolymer. The only difference is the total disappearance of the band located around 992 cm$^{-1}$. This can be justified by the dehydroxylation of illite at a calcination temperature equal to 950 °C. The band located at 1063 cm$^{-1}$ is attributed to the new anorthite phase, which seems to be weaker in quantity than the one formed in the GA$_1$$^{750}$ geopolymer, so we notice the appearance of the new bands in the GA$_1$$^{950}$ geopolymer located around 524 cm$^{-1}$ and 573 cm$^{-1}$ which are attributed to the vibration mode of the P–O bonds in the brushite (CaHPO$_4$·2H$_2$O) [51] already observed by XRD. In the same way, we observe the persistence of the two bands observed towards 3373 cm$^{-1}$ and 1641 cm$^{-1}$; they are attributed to the modes of vibrations, elongation, and deformation of the groups (O–H) coming from the phosphoric acid [48] and to the vibrations of the water molecules (H–O–H), respectively.

### 3.2.3. SEM Analysis of Geopolymeric Materials GA$_1$$^{550}$, GA$_1$$^{750}$, and GA$_1$$^{950}$ after 7 Curing Days

Scanning electron microscope observations of the synthesized geopolymers as a function of calcination temperature (550 °C, 750 °C, and 950 °C) are shown in Figure 16.

The examination of the SEM photos of the various geopolymeric materials highlights a consolidated and microporous microstructure, whatever the form of calcite and the temperature of raw clay calcination. However, this porosity decreases remarkably after the thermal decomposition of calcite. In fact, the comparison of Figure 16a–c shows that the microstructure of the geopolymeric material GA$_1$$^{550}$ and GA$_1$$^{750}$ are relatively finer and more porous than that of GA$_1$$^{950}$. It also seems that the amorphous phase formed during the geopolymerization is relatively more developed before the calcite decomposition. This observation confirms the results of the XRD analysis.

Thus, the presence of carbonates in the starting clay can influence the microstructure of the obtained geopolymeric materials, and mainly their porosity.

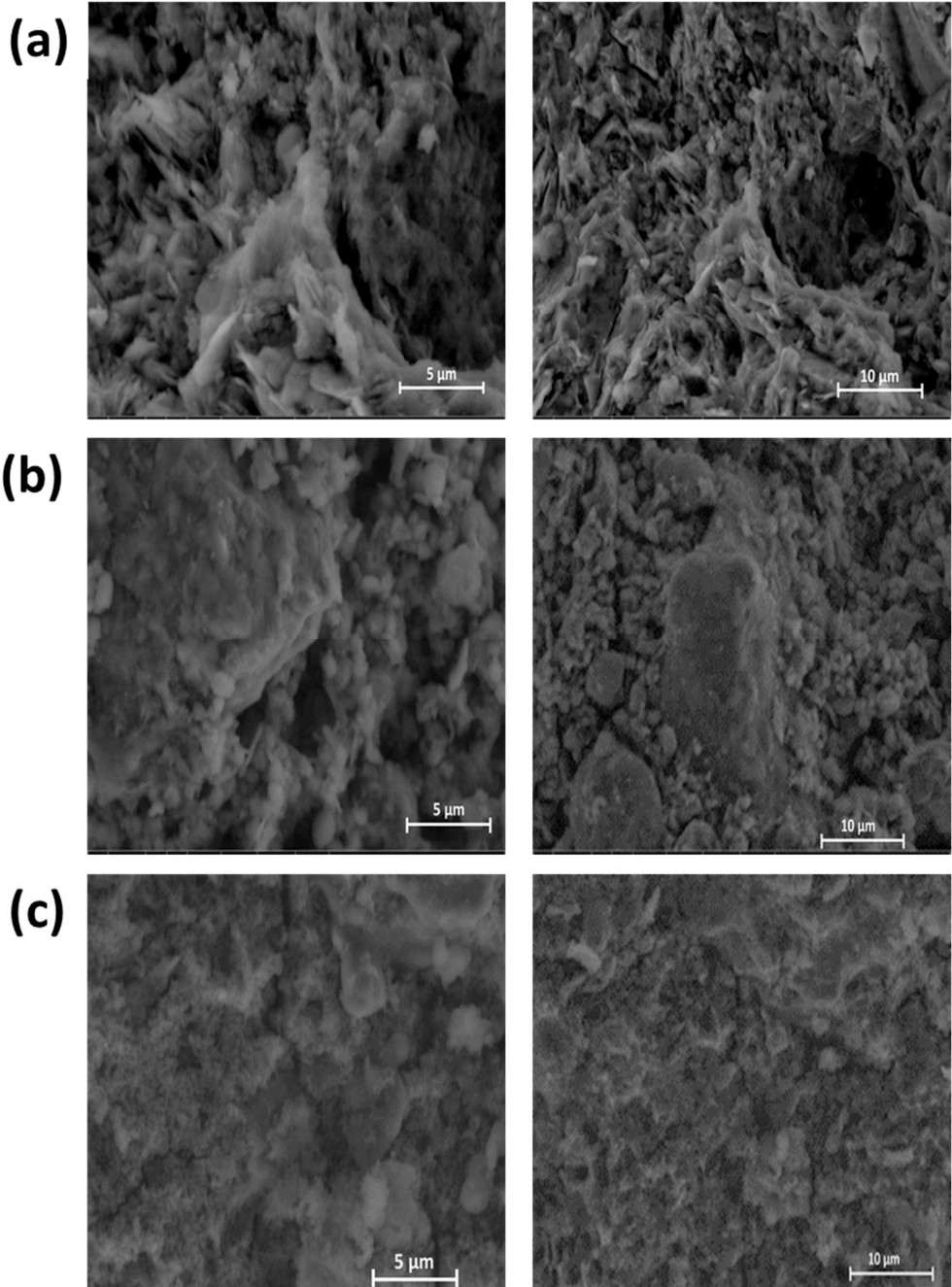

**Figure 16.** SEM analysis of geopolymeric materials: (**a**) $GA_1^{550}$, (**b**) $GA_1^{750}$, and (**c**) $GA_1^{950}$.

3.2.4. Mechanical Strength and Open Porosity of Geopolymeric Materials: $GA_1^{550}$, $GA_1^{750}$, and $GA_1^{950}$

Figure 17 presents the evolution of the diametric compressive strength and the open porosity of geopolymeric materials obtained as a function of the calcination temperature of Tunisian clay. The results obtained on geopolymeric materials synthesized from the raw A25 clay $GA_1^{25}$ and from a relatively pure metakaolin $GK_1^{750}$ [10] are also presented for comparison.

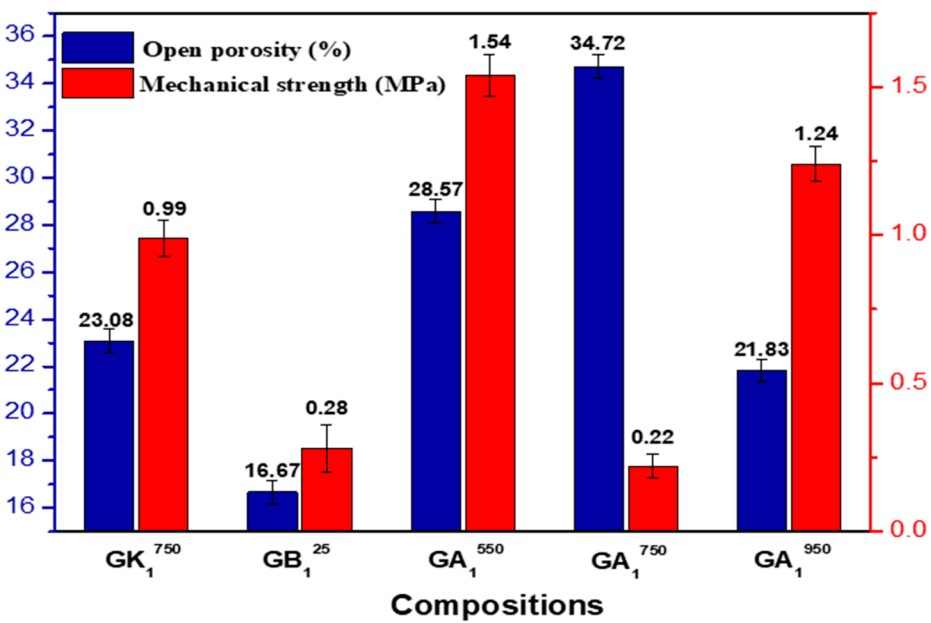

**Figure 17.** Evolution of the mechanical strength and the open porosity of the synthesized geopolymeric materials according to the calcination temperatures of the clay.

The examination of these results shows that the highest mechanical strength corresponds to that of the $GA_1^{550}$ sample (despite the presence of high open porosity), and the lowest one corresponds to that of $GA_1^{750}$. By linking these results with those of the previous analyses, we can deduce the following:

- The high mechanical strength (1.54 MPa) of $GA_1^{550}$ can be explained by the reinforcement effect of the persistent crystalline phases, such as quartz and illite, and the newly formed crystalline phases, even in small quantities, such as anorthite and aluminum phosphate. This effect explains, in particular, the deviation from the mechanical strength of the geopolymeric material synthesized from metakaolin ($GK_1^{750}$), where these phases are not present apart from aluminum phosphate. This effect also shows the importance of the presence of crystalline phases for improving the mechanical properties of geopolymeric materials. The high open porosity of the $GA_1^{550}$ compound is probably explained by the large quantity of gas ($CO_2$) released following the reaction between the calcite, still present at this temperature, and the phosphoric acid.

- The drop in mechanical strength (0.22 MPa) and the increase in open porosity of the $GA_1^{750}$ sample are probably explained by the formation of a small amount of the geopolymeric phase in this material. Indeed, at a temperature of 750 °C, the calcite is transformed almost entirely into lime CaO which can react with a part of aluminosilicates such as metakaolin to form new calcium aluminosilicate compounds, which greatly reduces the number of reactive phases (metakaolinite) responsible to react with phosphoric acid.

- In the case of the geopolymeric material $GA_1^{950}$, we note that the increase in the calcination temperature of the clay up to 950 °C generates an improvement in the mechanical strength of this material (of the order of 1.24 MPa). This increase can be explained on the one hand by the formation of the brushite phase, which plays the role of reinforcement [52], and on the other hand by the contribution of illite, dehydroxylated at this temperature, to the geopolymerization reaction and, consequently, the development of the geopolymeric phase. This also explains the relative decrease in the open porosity of $GA_1^{950}$.

- The mechanical strength of the geopolymeric material obtained from the natural raw clay $GA_1^{25}$ is relatively low (about 0.28 MPa). This is expected given the low reactivity of the raw clay toward phosphoric acid.

These data corroborate the results deduced from the other analyses (XRD, FTIR, and SEM). The mechanical strength, as well as the open porosity of the geopolymeric material, seem to be directly related to the type and quantity of the phases present in this material.

On the other hand, according to the results obtained, the effect of calcite, present in the starting clay, on the properties of the synthesized geopolymeric materials seems to be very important. This effect depends on the calcination temperature of the clay. For calcination temperatures lower than the decarbonation temperature, this phase remains relatively inert during the geopolymerization reaction. Under these conditions, the properties of the geopolymeric materials can be improved due to the reinforcing effect of the residual calcite or the lime formed following the attack of the calcite by the phosphoric acid and the formed anorthite. For calcination temperatures higher than the decarbonation temperature, calcite is transformed into relatively reactive lime (CaO), which can react, at these temperatures, with aluminosilicates (precursors of the geopolymerization reaction) such as metakaolin, to form other calcic aluminosilicate compounds, and consequently inhibits the formation of the geopolymeric phase in the synthesized geopolymeric materials, which significantly reduces their properties.

## 4. Conclusions

This study dedicated to the characterization of phosphoric acid-based geopolymers and Tunisian carbonated clay calcined at three different temperatures (550 °C, 750 °C, and 950 °C), allows us to draw the following conclusions:

- The calcination of the clay A25, mainly rich in quartz, kaolinite, illite, calcite, and clinochlore, causes the disappearance and transformation of different minerals, which have a great influence on the geopolymeric material obtained.
- Quartz persists in the geopolymeric materials, whatever the calcination temperature of the clay. It is inert against phosphoric acid. However, its presence in the mineralogical composition of the starting clay plays the role of reinforcement and generally ameliorates the mechanical properties of the obtained material.
- For geopolymeric material $GA_1^{550}$, the outstanding observed phenomenon was the decomposition of the calcite phase by the acid attack to form with the aluminosilicates the new anorthite phase ($CaAl_2Si_2O_8$). In addition, we note the appearance of an important amount of amorphous geopolymeric phase, which is more developed than the other geopolymeric materials ($GA_1^{750}$ and $GA_1^{950}$), as well as the appearance of the aluminum phosphate following the geopolymerization reaction. The high mechanical strength (1.54 MPa), which characterized the corresponding material, can be explained by the reinforcing effect of the present crystalline phases (persistent phases such as quartz and illite) and the phases formed even in weak quantities, such as anorthite and aluminum phosphate.
- For geopolymeric material $GA_1^{750}$, we notice the appearance of relatively important crystalline phases: one of aluminum phosphate, which is one of the products of the geopolymerization reaction, and the other of anorthite explained by the reaction between lime, formed by total decarbonation of calcite at 750 °C and a part of the aluminosilicates. Moreover, we notice the persistence of illite at this temperature. The low mechanical strength and the high open porosity can be explained by the low formed quantity of geopolymeric phases.
- For geopolymeric material $GA_1^{950}$, we observe the appearance of an aluminum phosphate crystalline phase, which is one of the products of the geopolymerization reaction, and the appearance of an anorthite crystalline phase explained by the reaction between lime and part of the aluminosilicates present. The quantity of these phases seems to be lower than that formed in the $GA_1^{750}$ geopolymer. This decrease is justified by the formation of a new crystalline phosphate phase which is brushite ($CaHPO_4 \cdot 2H_2O$). Besides the contribution of illite, the influence of calcite could be proven by the formation of brushite, which reinforced the structure of $GA_1^{950}$ and improved the mechanical strength.

The three calcination temperatures of the natural carbonated clay significantly modify the effect of calcite during the geopolymerization reaction and, consequently, on the properties of the obtained geopolymeric materials.

**Author Contributions:** Conceptualization, R.D., M.Z. and S.B.; methodology, R.D., M.Z. and S.B.; software, R.D.; validation, M.Z. and S.B.; formal analysis, R.D.; investigation, R.D.; resources, R.D., M.Z. and S.B.; data curation, R.D.; writing—original draft preparation, R.D.; writing—review and editing, R.D., M.Z. and S.B.; visualization, R.D., M.Z. and S.B.; supervision, S.B.; project administration, S.B.; funding acquisition, R.D., M.Z. and S.B. All authors have read and agreed to the published version of the manuscript.

**Funding:** This research received no external funding.

**Data Availability Statement:** Not applicable.

**Acknowledgments:** The authors would like to acknowledge Imerys France for providing us with kaolin and would like to thank the Materials Chemistry and Catalysis Laboratory, ISSBAT, Tunis El Manar University, for facilitating the performance of the XRD analyses.

**Conflicts of Interest:** The authors declare no conflict of interest.

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
