# Peer review of "Study of Carbonated Clay-Based Phosphate Geopolymer: Effect of Calcite and Calcination Temperature"

_minerals, doi:10.3390/min13020284_

Round 1

Reviewer 1 Report

see my annotation

Some figures need to be improved

Reviewer 2 Report

This work is interesting and the topic is worthy of research. Here are some minor comments:

1) Please improve the literature review regarding the role of calcium in phosphate-based geopolymers. Here are some suggestions:

10.3389/fmats.2019.00106

10.1016/j.cemconres.2019.105932

10.1016/j.cemconres.2022.106840

2) consider discussing the xrd results of the raw clay in the material section to introduce the crystalline phases to the readers.

3) Please explain more the L/S ratios. Is the solid content available in the phosphoric acid included in the solid fraction?

4) If possible, merge figures 6-8 in one for better comparison.

Reviewer 3 Report

Report on “Study of carbonated clay-based phosphate geopolymer: effect of calcite and calcination temperature”, Derouiche et al.
General comments.
This manuscript reports an extensive series of well-executed experiments on a geopolymeric material based on an impure Tunisian clay. However, the compressive strengths of these materials (1.24 - 1.54 MPa) are extremely low compared with other reported phosphate-metakaolinite geopolymers (e.g. 36.4 - 93.8 MPa, Applied Clay Science 147 (2017) 184). The authors should comment on the weak strength of their material and justify their study with suggestions for potential practical applications of their geopolymers.
They should also point out in their introduction section (line 44) that although acid-based phosphate geopolymers are the most widely reported, the formation of alkali-based geopolymers in which the tetrahedral silicate groups are partially replaced by phosphate groups has also been reported (Ceramic Transactions 145 (2006) 187-99).
Particular points for attention.
Line 51. An explanation is required of the difference between kaolinite and “natural clay”. Is this simply a matter of the presence of impurities in natural clay, or is some other difference implied?
Line 80. Is the presence of free silica and/or 2:1 clay such as illite supported by XRD? If so, a reference to this should be included here.
Line 98. For consistency the experimental DTA/DTG details (heating rate, atmosphere) should go in here (section 2.3) with the other experimental details.
Line 134. Were the FTIR samples suspended in KBr or Nujol? This experimental detail should be stated.
Line 140. Were the SEM samples in the form of broken or polished pieces? Details of the coating (if any) and the instrument working conditions chould also be given.
Line 163. Was the goethite phase postulated here detected by XRD or FTIR? This phase is not mentioned in Section 3.1.4 or 3.1.5, or in Table 2. If this is merely a suggestion which is unsupported by experimental observation, this should be clearly stated.
Line 178 ff. Are all these suggested reactions consistent with the XRD results? If so, this should be stated.
Table 2. spelling- kaolinite. Also, should the heading to Table 2 be “A summary of the crystalline phases present in the different compounds”?
Figure 14. This is too dark to be useful. The contrast should be improved.
Lines 436 and 497. Is “mechanical resistance” the same as “compressive strength”? If not, the term “mechanical resistance” should be defined, but if these terms are the same the more commonly understood term “compressive strength” should be used throughout.
Figure 15 Error bars are required in this figure.
If all these points are satisfactorily attended to, the paper would be suitable for publication.

Reviewer 4 Report

In this paper,author aims to use natural carbonated Tunisian clay as an aluminosilicate precursor for the elaboration of phosphate-based geopolymersï¼›The calcination temperature of the natural carbonated Tunisian clay modify significantly the effect of calcite during the geopolymerization reaction and consequently on the properties of the obtained geopolymeric materials.

1. In Abstract,CaHPO4.2H2Oshould changed to CaHPO4·2H2O.

2. In Section2.3 Methods,the author introduced the experimental methods. I think it is better to add a workflow chart and introduce the experimental methods in the order of experiments, which is more convenient for reading.

3. In Section3.1.2. Thermal behavior of raw clay ,the reason for choosing 950。C as calcination temperature needs to be modified.

4. In Section3.2.2. FTIR spectra of geopolymeric materials: GA1550, GA1750 and GA1950 , the conclusion mentioned by the author "Thus, we can note the chemical inertia of these phases with respect to the phosphoric acid at a temperature of calcination which is equal to 550 °C. " lacks supporting evidence. It is suggested that the author further explain the principle of this conclusion or add references to explain it.

5. In SectionSEM analysis of geopolymeric materials GA1550, GA1750 and GA1950 after 7 curing days   , author should clarify the influence of carbonates on the microstructure of the obtained geopolymeric materials.

6. In Conclusions,at the process of analyzing the formation of geological polymer GA1950, the explanation of the influence of calcite on the properties of geological polymer GA1950 should be added.

7. In References,the format of [6]/[45]/[50] needs to be modified.

Suggestion: major revision
